# A Methodology for Assessing the Implementation Potential for Retrofitted and Multifunctional Urban Green Infrastructure in Public Areas of the Global South

Tanja Fluhrer [1], Fernando Chapa [2] and Jochen Hack [2,*]

1   Department of Civil and Environmental Engineering, Technical University of Darmstadt,
    64289 Darmstadt, Germany; tanja.fluhrer@stud.tu-darmstadt.de
2   Research Group SEE-URBAN-WATER, Section of Ecological Engineering, Institute of Applied Geosciences,
    Technical University of Darmstadt, 64289 Darmstadt, Germany; chapa@geo.tu-darmstadt.de
*   Correspondence: contact@geo.tu-darmstadt.de

**Abstract:** Urban green infrastructure (UGI) provides multiple functions that combine ecological and social benefits. UGI is being increasingly promoted and implemented in the Global North. In other parts of the world, such as in the Global South, infrastructures for UGI implementation and promotion are sparse. The state of infrastructure development and informal settlements in the Global South present different constraints and demands that should be explicitly addressed. This study presents an approach to addressing the specific conditions and physical limitations of UGI development in urban areas of the Global South. A four-step methodology was developed to assess the implementation potential for retrofitted and multifunctional urban green infrastructure in public areas. This methodology consists of (1) an initial site analysis, (2) defining design criteria and general strategies, (3) exploring the different dimensions of multifunctionality as the basis for deriving spatial typologies, and (4) assessing spatial suitability for potential placements for UGI elements. The methodology was applied to a study area in the metropolitan region of San José, Costa Rica. The results indicate the potential to improve the hydrological (up to 34% of surface runoff reduction), ecological (an increase of green space by 2.2%, creation of 1500 m length of roadside greenery and two new habitat types), and social conditions (2200 m of road type upgrading) of the site through UGIs. This assessment of different multifunctionality dimensions can serve as a guide for future UGI promotion and implementation in urban areas of the Global South.

**Keywords:** green infrastructure; urban; multifunctionality; retrofitting; sustainability; neighborhood level; Costa Rica

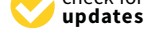



## 1. Introduction

Nature-based solutions (NbS are measures adapting or based on natural processes that address societal challenges. NbS have been identified as a promising approach to deal with socio-ecological problems and are being considered in urban contexts [1]. In Europe, for instance, with NATURVATION (NATure-based URban innovation), the European Commission is funding a four-year project (2017–2020) involving 14 institutions across Europe to investigate the potential of NbS for responding to urban sustainability challenges [2,3]. In urban contexts within the USA, NbS are promoted as green infrastructure country-wide through a special program of the United States Environmental Protection Agency (EPA) based on the Clean Water Act [4]. As a result, several policies and technical guidelines for promoting and implementing green infrastructures (GI) have been produced for the USA. Similar guidelines have been established in Australia through water-sensitive urban design (WSUD). The federal, state, and territory governments of Australia agreed upon the National Water Initiative (NWI) in 2004 and 2006 to "Create Water-Sensitive Australian Cities", encouraging the adoption of the WSUD approach [5]. Sustainable drainage systems (SUDS) in the UK are another example guideline for development and promotion of this

kind. It shows that planning, promotion, and implementing guidelines for NbS in urban contexts exist, at least in the USA, Australia, and Europe. However, these applications focus on new urban developments or interventions in existing urban drainage structures and green spaces.

In the Global South, governments and public authorities have not yet adopted NbS. Urban development in those regions has occurred informally or has gone unplanned, resulting in urban areas without urban drainage and wastewater treatment systems and a lack of public or private green spaces [6]. This has led to environmental damages (e.g., biodiversity loss due to habitat loss and environmental contamination) [7] and insufficient amenities for social well-being [8]. The potential multifunctionality of urban green infrastructures (UGIs), as NbS in the context of urban areas, has been proven in numerous cities in the USA [9] and Europe [10]. It is highly probable that UGIs can also address these deficits in the Global South and serve as NbS. However, the lack of and/or poor state of public infrastructures limit the implementation of UGIs following pre-established standards. There is a need for studies that explicitly address the specific conditions and physical limitations of UGI development as well as the potential for multifunctional implementation. Considering multifunctionality in the development of UGIs may facilitate a comprehensive upgrading of residential areas in the Global South.

Multifunctionality is a principal characteristic of UGIs [9–11]. It means that UGIs—or more generally, green spaces—provide a variety of functions (e.g., ecological, social, economic; [10]). For instance, when neighbors organized street markets in neighborhoods with green spaces, the sense of belonging and the level of cooperation were greater than those of neighborhoods with no green space [12]. Especially in the application of UGIs, their hydrological function in the context of stormwater management plays an extraordinary role [13]. Citizen initiatives, supported by an enabling and stimulating governance style, can significantly contribute to social cohesion, institutional innovation, and diversity in urban green space management [14]. A decisive factor for the achievement of different functionalities is, above all, the specific design (geometries and spatial distribution) and placement (i.e., embedding of green infrastructure elements in space). In an urban retrofitting context, space is usually limited and predefined by competing single-purpose functions (e.g., vehicle traffic or housing) [15]. Since the UGI portfolio is diverse, but not all elements are locally suitable, it is important to define constraints and opportunities in an early stage of the design process to select the most suitable elements [16].

This study addresses the lack of research on planning multi-functional UGIs and practical guidelines by proposing a methodology that provides guidance for the placement, dimensioning, and spatial distribution of multifunctional UGI elements in an urban setting of the Global South. The application of the methodology takes place at a neighborhood scale in the metropolitan region of Costa Rica. The methodology builds on insights from existing UGI design guidelines and consists of four steps: (1) analysis of the site, (2) definition of design criteria and implementation strategies, (3) development of spatial typologies, and (4) spatial suitability assessment of UGI elements. The multifunctionality dimension of UGIs is considered from a hydrological, ecological, and social perspective, integrated as part of the design criteria and implementation strategies of step (2). This UGI implementation methodology is particularly innovative, as it explicitly aims for multiple functions of green elements to serve a variety of demands in a complex urban setting. When considering public space, synergies in functions of the mobility sector are potential additional benefits.

Our results show in detail the explicit spatial placement potential for different kinds of retrofitted UGI elements of suitable geometries within a dense urban fabric and the multifunctionality that could be achieved in each of the considered dimensions. This allows for the implementation of retrofitted UGIs adapted to the specific conditions and constraints of urban areas in the Global South, such as constraints due to vehicle traffic and driveways, predominant road designs, and the presence and spatial distribution of existing green features within the urban matrix. We conclude that the proposed methodology provides a suitable basis for the development of placement strategies (geometries of prototypes,

spatial distribution at the neighborhood scale) for local planners to achieve a high degree of multifunctionality and to promote UGI elements as valuable integral components of urban planning.

We propose a general methodology to effectively improve multifunctionality in an urban residential retrofitting setting through a specific UGI placement strategy.

## 2. Materials and Methods

### 2.1. Methodology to Assess the Placement Potential of Multifunctional Urban Green Infrastructure

The achievement of a high degree of multifunctionality is intended through a coherent placement and spatial distribution strategy that facilitates the creation of networks and increases connectivity by interlinking existing green spaces functionally and physically with newly placed UGI elements at a neighborhood scale. Thereby, integration and combination of existing green spaces and UGI elements with other urban infrastructures, such as transport infrastructure, drainage systems, and buildings, are pursued. Instead of a blanket one-size-fits-all solution, the methodology enables contextual analysis by incorporating information about the limitations or constraints of a site in implementing multifunctional UGI, and the harnessing of collected data to identify landscape potential and to maximize it based on empirically revealed conditions.

The specific assembly, properties, and spatial distribution—such as the areas occupied by houses, sidewalks, streets, green public or private spaces, and the use of these spaces—reveal specific opportunities for multifunctional UGIs and potential constraints. Therefore, the spatial configurations of these site features are analyzed first in a methodological step (1) as the basis for multifunctional UGI implementation. This evaluation of site characteristics refers to important elements of the landscape: configuration of particular urban elements and topography as well as the spatial distribution of properties, land uses, green spaces, and drainage systems. In step (2), the design criteria and placement strategies to achieve different dimensions of multifunctionality are defined. In this step, hypotheses, priorities, and possible decisions for the design process are formulated, supported by the findings of (1). Due to the multifunctional character of UGIs, many different design criteria, i.e., priorities, can be set; for example, stormwater management, biodiversity enhancement, urban heat island reduction, air quality improvement, social cohesion, recreation, and/or education. Strategies specify the defined design criteria and concretize their specific implementation. In a third step (3), spatial typologies for different UGIs are formulated considering a portfolio of suitable locations based on the site analysis (1). Various scenarios can be proposed by considering the characteristics of open green spaces or the dominant road network, such as street dimensions and traffic volume. Depending on which information is available, further criteria can be added. This step includes a review of possible UGI elements to understand their functions and select the most suitable ones for the particular area. Limitations for dimensioning and placement are given by the site characteristics (1), spatial typologies (3), and design parameters recommended for individual UGI elements (e.g., the maximum attached drainage area). For instance, the minimum space requirements of specific elements can be determined depending on the desired design functionality target (e.g., hydrological performance) resulting from the selection of respective design storm events for their dimensioning. As result, design proposals are developed, including specific suggestions for placement, dimensions, and technical configurations, as part of step (4). Figure 1 illustrates the four-step methodology.

To illustrate the proposed methodology in more detail, its application is described in the context of our case study example.

### 2.1.1. Step 1—Analysis of the Site (Study Area)

The study area represents a typical highly urbanized residential area located in the Greater Metropolitan Area in the municipality of Flores, 13 km northwest of the capital, San José, in Costa Rica. Its area of 33 ha, with a population of about 2510, is mainly narrowly built with paved streets. Steady population growth over the past decades [17] and only a few

small green and recreational areas are key characteristics of the site, which result in limited available space for UGI implementation. The area represents a hydrologically closed drainage area within the urban watershed "Quebrada Seca-Río Burío" (see Figure 2).

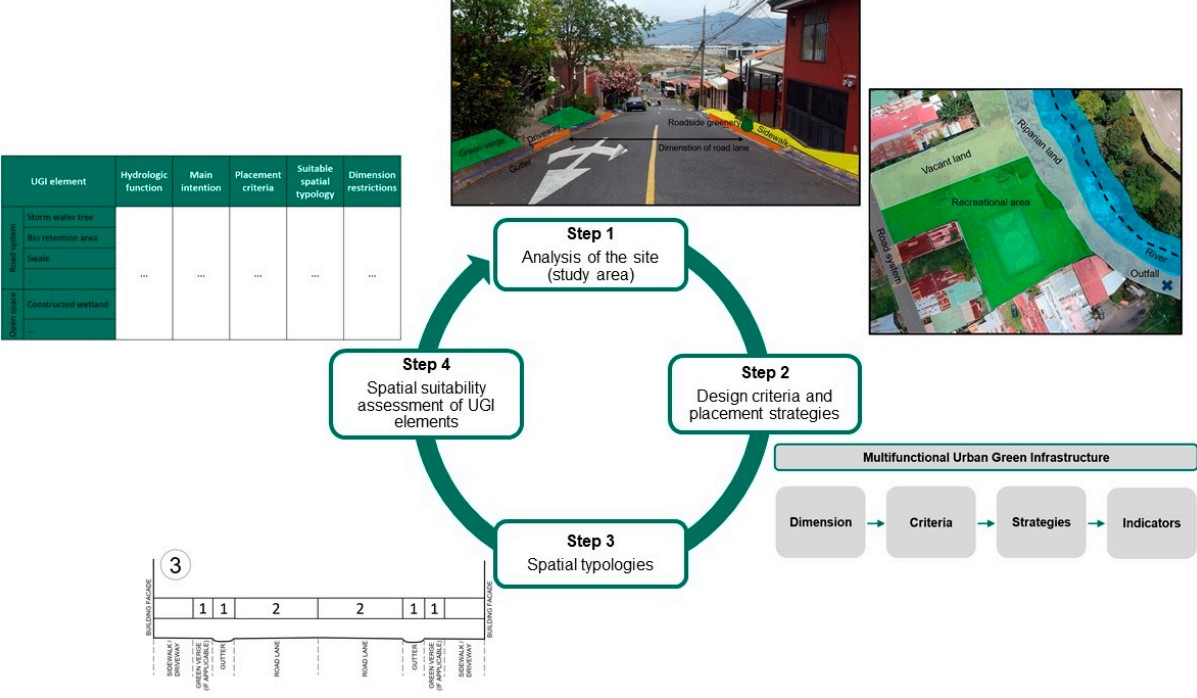

**Figure 1.** Methodological steps to assess the placement potential of multifunctional urban green infrastructures (UGIs).

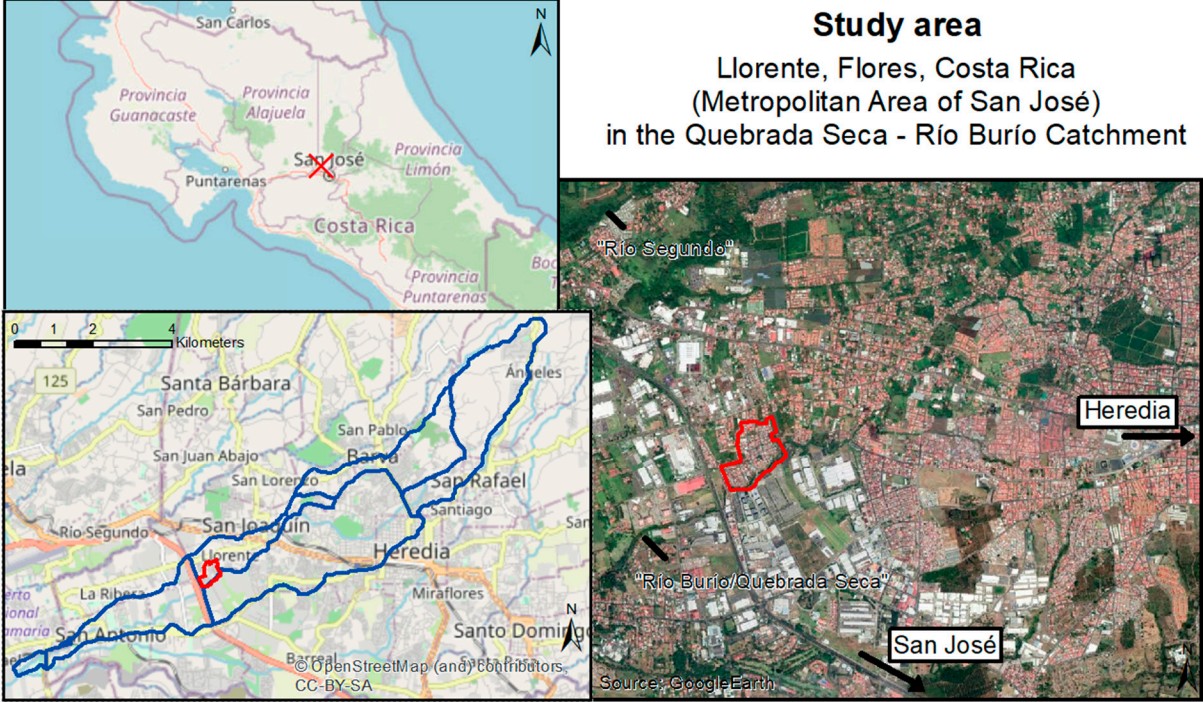

**Figure 2.** Map of the larger Quebrada Seca-Burío Catchment (blue) and the study area (red) in the Metropolitan Area of San José, Costa Rica. Map sources: Open Street Maps and Google Earth.

Spatial information was gathered mostly through fieldwork (georeferenced recording and tracking of site features) of the Research Group SEE-URBAN-WATER (www.tu-darmstadt.de/see-urban-water) from April to August 2019. Additional information (i.e., cadastral map, street network, property situation) was provided by the local municipality. A drone was used to retrieve aerial information in video format. The data were digitalized in Geographic Information Systems (GIS) for further analysis.

Existing Green Network and Social Accessibility

The study area is a residential zone, with around 60% of the area covered by one- to two-floor single-family houses. The few existing green spaces, mainly located along the river, were categorized as recreational areas (playgrounds and sports facilities), vacant land (in public hands), undeveloped properties (in private hands), riparian land, and unsealed areas (Figure 3). The latter represent areas that belong to the Municipality and are reserved for future development. Along streets, a network of green verges and roadside greenery exists. We considered the social accessibility of recreational areas based on the proximity (distance and time to reach a recreational green space) and aerial extent (size). International standards vary here in their suggestions: The World Health Organization (WHO) suggests at least one 2 ha large green area no more than 300 m long at a walking time of 5 min or less from home [18], whereas the "European Common Indicator" only suggests a minimum of 0.5 ha of public green area within 300 m [19].

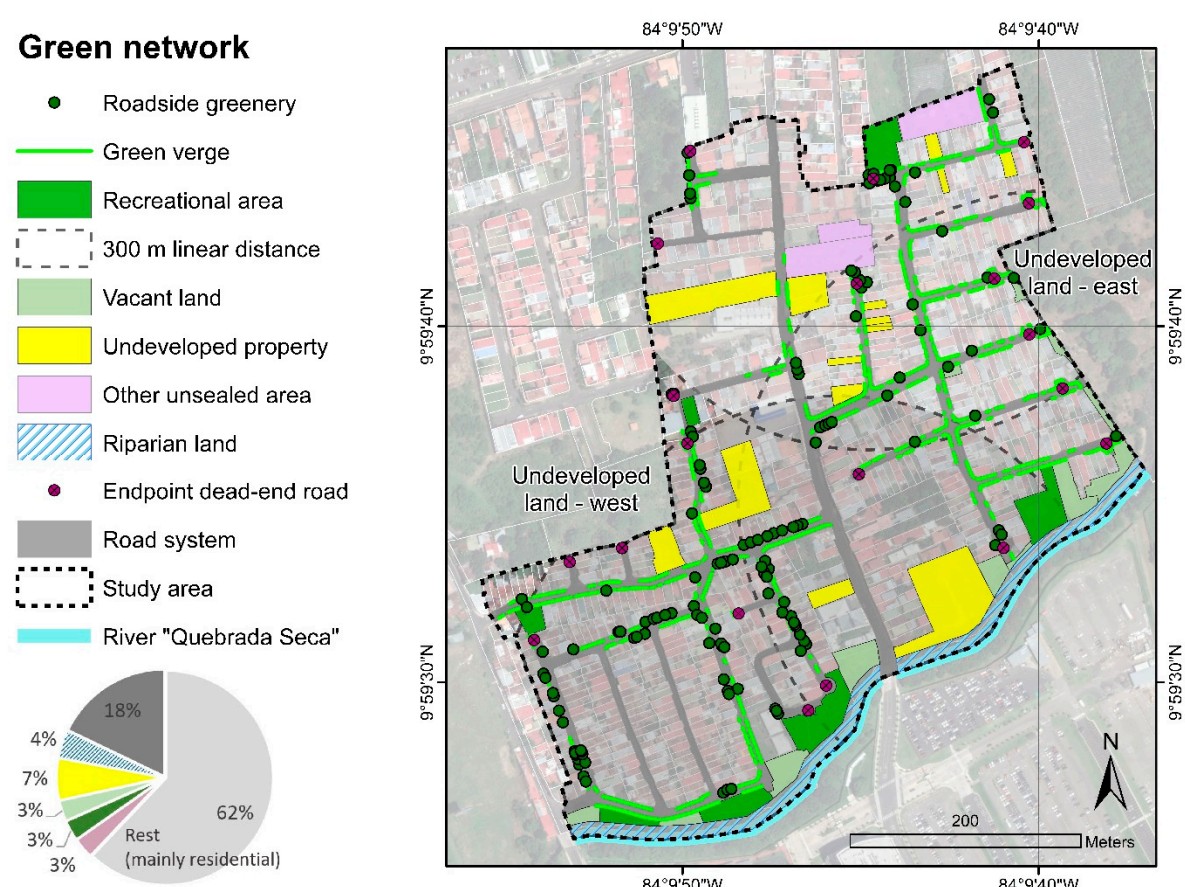

**Figure 3.** Map of the study area illustrating the existing network of green elements, land use distribution, and 300 m distance limits for recreational areas (source of the background satellite image: Google Earth).

While in the study area, the three largest recreational areas can be reached by almost all residents who are within a 300 m distance (see Figure 3) and within the limits of the study area, access is mainly limited to highly trafficked or dead-end roads. Hence, the walking distance can exceed 5 min from remote parts of the area. In addition, the largest recreational

green space of 2838 m$^2$ does not meet the suggested 0.5 ha. Green space recommendations indicate 5 m$^2$ per habitant [19]. In our case, only 10,960 m$^2$ of public recreational green space are available; 12,550 m$^2$ would be needed to meet the recommendation. Thus, the dispersed, too small, and hardly accessible public green spaces of the study area are a clear deficit in social functioning, but also in ecological and hydrological functioning. Figure 3 illustrates the spatial distribution in quantity and type of existing green and unbuilt spaces as well as their accessibility.

Existing Drainage System

Domestic graywater and stormwater from sealed surfaces and roofs are routed through open gutters along the streets into inlets to the sewer system. From there, the runoff is directly drained, without any treatment, into the river. This kind of storm and wastewater handling is typical in the entire Greater Metropolitan Area. It promotes a fast conveyance of high amounts of stormwater and untreated graywater into water bodies without consideration of other ecological, hydrological, or social functions of the drainage areas. Based on the 10 outfalls to the river, the study area can further be divided into 11 sub-basins, including one draining into a neighboring area, varying in size from 0.7 to 9.6 ha. Figure 4 illustrates the spatial delimitation of individual drainage networks in size, discharge directions, number of connected units, and outfalls to the river.

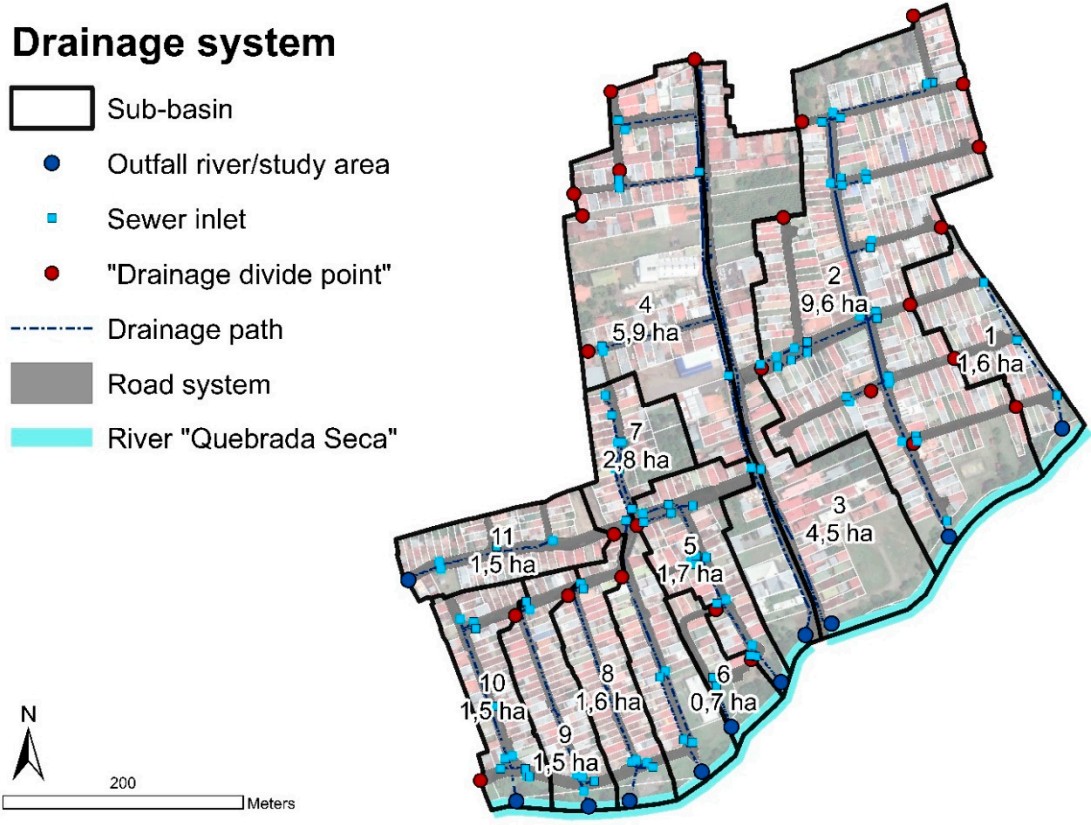

**Figure 4.** Map of the study area's storm water drainage system, including the sub-basin, drainage paths, and dividing points (source of the background satellite image: Google Earth).

Road Typology and Hierarchy

The area has a paved road network focused on motorized traffic. The uniform profile typically consists of two road lanes, sidewalks on both sides that are frequently interrupted by driveways entering the properties for parking, green verges, and gutters for the open discharge of runoff. Speed humps and on-street parking can be found to slow down the traffic speed. Cross-sections were measured at nine different locations to subsequently

classify the roads into four hierarchical types: access roads (connecting a neighborhood with the main road), local roads (roads conveying traffic through a neighborhood), residential roads (roads that primarily serve as access to residents' properties), and small residential roads (functioning as residential roads, but with reduced cross-sections). According to this classification, most streets that are residential are dead-end roads.

Based on traffic counting and field observations, roads were further classified according to their traffic volume relative to the traffic in the entire study area. The traffic counting took place on June 2019 (Tuesday) as a representative weekday from 6:00 till 18:00 at an intersection with particular relevance (connecting main roads with external traffic and two dead-end roads). The flow of traffic and vehicle patterns were manually documented by project members and simultaneously recorded. Hence, very high traffic volume is estimated for the only direct through road. High traffic volumes (>250 cars per hour counted) are expected on roads connecting to the free trade zone and medium traffic volume for further connecting roads. Low traffic volume was assumed for all dead-end roads and small residential roads.

Resulting from the combination of road hierarchy and traffic volume, different road types (1–5) are defined. Figure 5 illustrates the spatial distribution of road types (categories) based on road hierarchies and traffic volumes. The road type is directly linked to the available space for and the specific type of UGI. Due to the topography, all roads with North–South direction have a steep slope, draining into the river in the South.

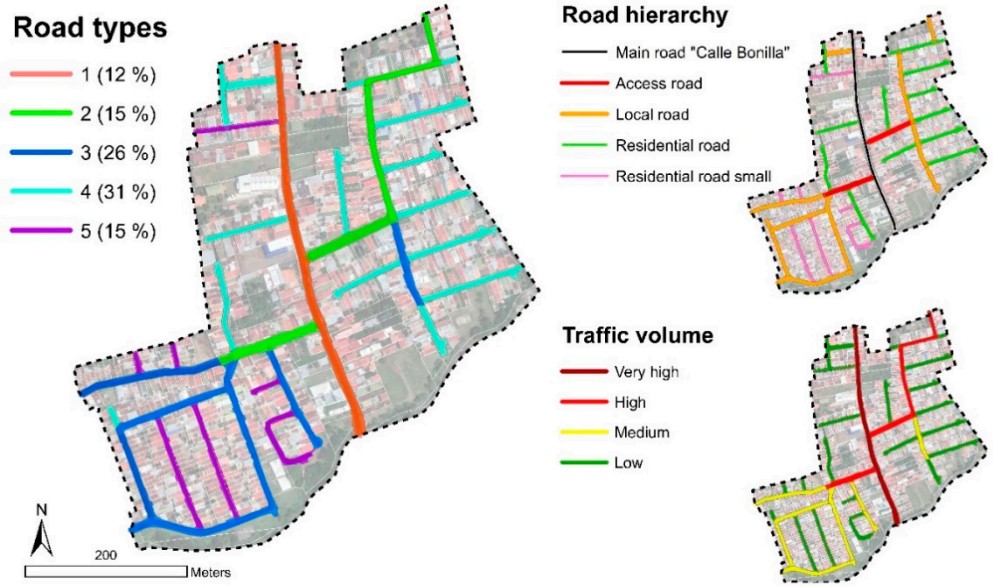

**Figure 5.** Spatial distribution of road types (1–5), road hierarchies, and traffic volumes within the study area (source of the background satellite image: Google Earth).

### 2.1.2. Step 2—Design Criteria and Placement Strategies

To develop a feasible methodology, we prioritized functionality related to the three dimensions considered in this study (i.e., hydrological, ecological, and social dimensions) using the respective design criteria and strategies for dimension-related improvements (see Figure 6). In our methodological understanding, criteria define "what" the UGI placement and design should achieve, and the strategies describe "how they should be achieved" or "by which means". A vision unites the strategies and can be formulated as: "Linking green spaces through multifunctional urban green infrastructures in roads and open spaces to create an interconnected network that reduces surface runoff, provides opportunities for social activities, increases safe accessibility within the neighborhood, and enhances the biodiversity of the site."

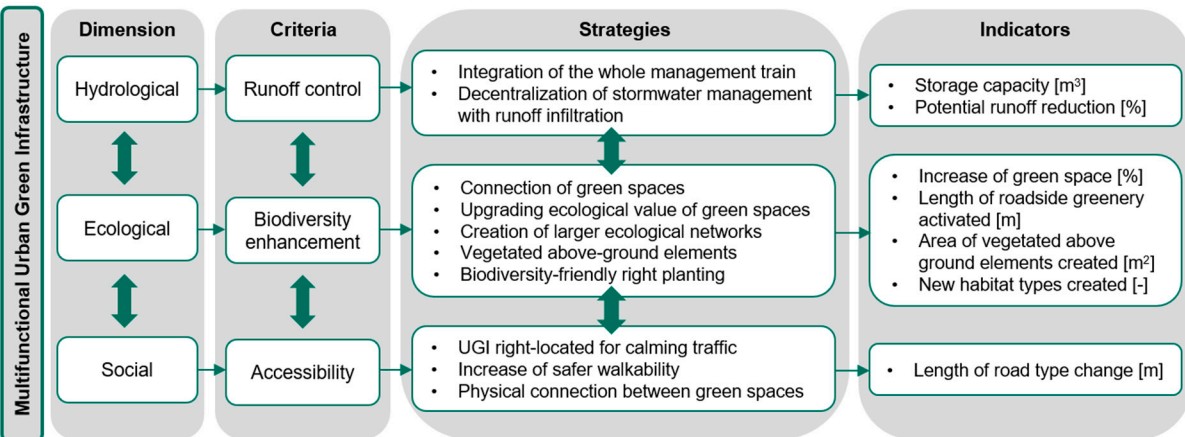

**Figure 6.** Dimensions, criteria, and strategies for multifunctional green infrastructures used in this study.

The prioritized criteria for the three multifunctional dimensions in this study were selected based on expressed concerns (flooding, biodiversity loss) and demands (lack of green spaces) from stakeholders in the study area. The prioritized criterion for the hydrological dimension was runoff control. It aims to mimic the natural water cycle by reducing stormwater generation through infiltration and retention to increase evapotranspiration. A side effect of achieving these aims is a minimization of water pollution through contaminated urban runoff discharging into urban streams. A preliminary quantitative runoff control focus was chosen because of the significant problems of urban flooding encountered in densely urbanized tropical regions [15] and the specific capacity of UGIs in addressing them [20]. The strategy to accomplish the criterion of runoff control was to increase runoff infiltration to decentralize the stormwater management in the urban landscape and to integrate the whole management train of UGIs in the urban landscape. To evaluate this criterion, design proposals were assessed regarding the [0–100%] increase of runoff storage capacity and the number of elements of different parts of the management train (source control, conveyance, end control; [0–3]) considered.

Within the ecological dimension, the prioritized criterion was biodiversity promotion. Hence, the implementation of UGI elements aims to increase biodiversity by increasing connectivity and upgrading the ecological value of green spaces. It is assumed that, by connecting fragmented habitats and creating larger ecological networks, as well as new habitat types (e.g., through wetlands), through the placement of ecologically functioning UGIs of additional ecological value, biodiversity can be increased and strengthened [21]. A particular opportunity to increase the degree of connectivity is found in riparian corridors with remaining natural conditions (e.g., vegetated river banks, natural river bed and course) [22,23]. The ecological dimension (criterion of biodiversity enhancement) is evaluated regarding the relative increase of green space, length of roadside greenery, area of above-ground vegetated elements, and area of new habitat types created by the UGI proposals.

In our study, accessibility is the prioritized criterion for the social functionality dimension. It is related to the placement and design of UGIs to calm traffic for better and safer walkability, as well to increase the physical connection to green spaces for recreational activities by encouraging outdoor activities within a neighborhood [24]. The criterion of accessibility was evaluated based on the length of roads, with a category for the improvement of roads, increased by the UGI proposal.

### 2.1.3. Step 3—Spatial Typologies

The target of this step is a portfolio of suitable public locations within the neighborhood. Defined spatial typologies help to limit possible placements of UGI elements within the site. Depending on the defined road typologies in Step 1, various scenarios and priorities can be set to define the most suitable locations for implementation within the existing road system and open spaces. For each road type, the first- and second-ranked locations

for UGI implementation were determined. Primarily, UGI elements were considered where the main functions of the existing infrastructure would be unaffected and enough space was simultaneously available to prevent conflicts. As a precondition, the main road of the neighborhood was not to be restricted any further due to tight space conditions and the highest traffic volume (see road type 1). In other roads with a high volume of traffic, the aim was not to establish UGIs in the road lanes. Preferred road components used for implementing UGI elements were gutters and green verges (see road types 2 and 3). If the dimensions and the traffic volume enabled it, implementation of UGI elements in the road lanes was desirable, offering a pedestrian-friendly design (see road types 4 and 5). An overview of the preferential locations of UGIs for different road types developed for this study is presented in Table 1 and Figure 7.

**Table 1.** Definition of road types based on traffic volume and hierarchical order, relative percentage of the total road network, and the preferred and second-ranked locations for UGIs.

| Road Hierarchy | Traffic Volume | Defined Road Type | Percentage of Total Road Network | Preferred Location | Second-Ranked Location |
|---|---|---|---|---|---|
| Main road | Very high | 1 | 13% | - | - |
| Access road Local road Residential road | High High High | 2 | 15% | Green verge, gutter | - |
| Local road Residential road | Medium Medium | 3 | 26% | Green verge, gutter | Road lane |
| Local road Residential road | Low Low | 4 | 31% | Road lane, green verge, gutter | Sidewalk |
| Res. road small | Low | 5 | 15% | Road lane, gutter | Sidewalk |

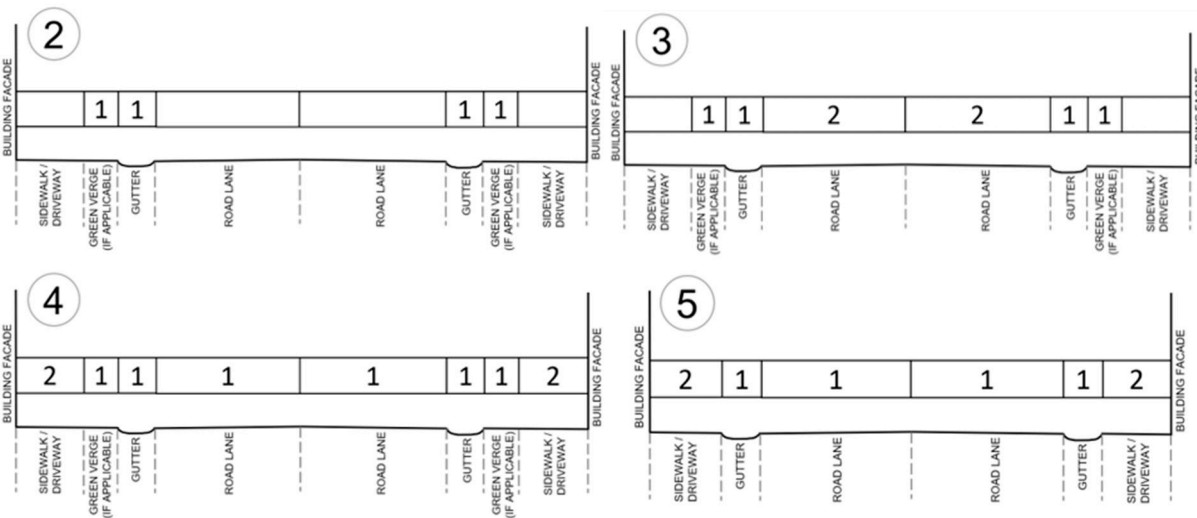

**Figure 7.** Cross-sections of road types 2–5 with indications of preferred and second-ranked location of UGI elements.

Regarding the implementation of UGIs in open spaces in the study area, areas of public green space were assumed to be suitable: recreational areas, vacant land, and riparian land (see Figure 3). Suitable UGI elements for each type of open space can be defined depending on their characteristics. According to this framework, recreational areas are most suitable for UGIs that allow multiple uses. Within the riparian land, elements connecting with the

blue network are preferable. Vacant land poses no special requirements. Table 2 shows which characteristics of UGIs are appropriate depending on the type of open space.

**Table 2.** Areas of different types of open spaces in the study area and their respective preferred UGI characteristics.

| Type of Open Space | Area (m$^2$) | Preferred UGI Characteristic |
|---|---|---|
| Recreational area | 10,960 | UGI elements allowing multi-use |
| Vacant land | 8770 | Imposes no special requirements on elements |
| Riparian land | 11,790 | Elements suitable for connecting with blue network |

Together with the usable existing road system, in total, around 24% of the study area is considered to be suitable for UGI placement (see Figure 8). Depending on which information is available for a site, further criteria (e.g., soil conditions, suitable spaces on properties and roof types) can be added.

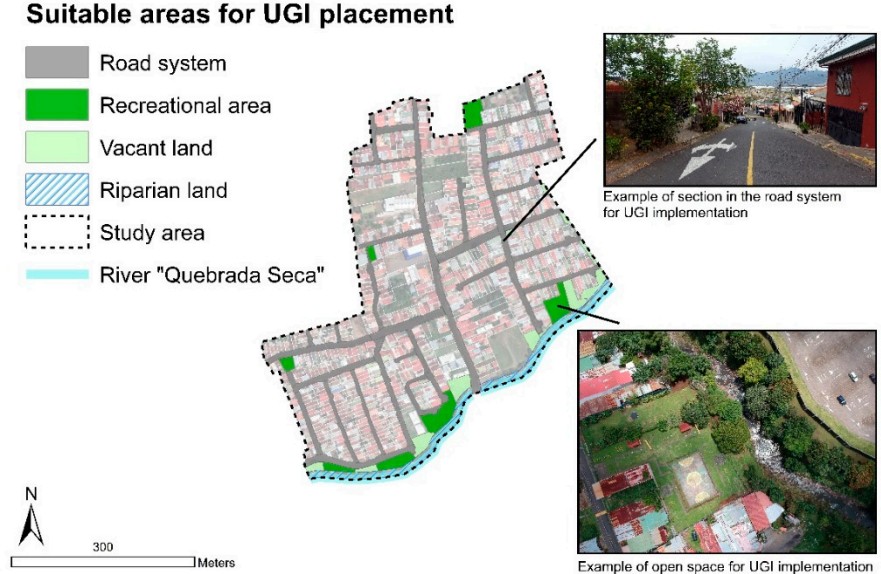

**Figure 8.** Distribution of suitable areas for UGI placement in the study area (source of images: Dennis Jöckel and Tanja Fluhrer).

### 2.1.4. Step 4—Spatial Suitability Assessment of UGI Elements

UGI elements can be categorized in several ways, such as their main functions, ecosystem services, components, or scale of application [25,26]. For implementing the most suitable types of UGI elements and creating a diverse network across a neighborhood, it is necessary to understand the many, often intertwined, functions of UGI elements. There are many features that provide important hydrological functions for controlling the frequency of runoff, flow rates, and volumes of runoff. Therefore, UGI elements for source control are located more upstream, whereas elements for surface runoff retention often have end-control positions and are located more downstream in the so-called management train [27].

Based on a review of 12 design guidelines [27–38] of five countries (United Kingdom, USA, Canada, Australia, and Singapore), common UGI elements that can be implemented in public space were selected for the study area. Table 3 shows the elements selected, which are categorized according to their hydrological functions and expected qualitative performance for approaching their potential in terms of functionality (runoff control for small and large events, ecology, social) and land use. Real performance is dependent on local conditions, the right design, and maintenance of UGI elements.

**Table 3.** Urban green infrastructure elements and their potential multifunctional performance based on information from [27–38] and qualitative classification by the authors; ++ = very high, + = high, o = medium, - = low/none.

| UGI Element | Runoff Control—Small Evens | Runoff control—Large Events | Ecology | Social | Land Use |
|---|---|---|---|---|---|
| Permeable pavement | + | - | o | + | ++ |
| Storm water tree | + | - | ++ | + | o |
| Bio-retention area | + | o | + | ++ | - |
| Swale | + | o | + | + | - |
| Infiltration trench | + | o | o | o | o |
| Detention basin | + | + | + | + | o |
| Retention basin | + | ++ | + | + | - |
| Constructed wetland | + | ++ | + | + | - |

As on-site features for stormwater management, UGI elements are often intended to control smaller and short- to medium-duration rainfall events. However, some systems with greater surfaces, such as retention basins and constructed wetlands, can also cater to larger storms. Others have poorer performances during long wet periods, such as permeable pavements and stormwater trees. In terms of ecological performance, especially elements with the possibility of intensive planting are classified as high. Larger vegetated UGI elements, such as basins and constructed wetlands, can therefore generally count on a higher value. In the social dimension, especially bio-retention areas designed as smaller systems can be integrated as traffic regulating measures along streets. No additional land is necessary by implementing permeable pavement, since multiple functions can be easily united in one location. Infiltration trenches only need minimal extra land when well designed, as small width and narrow shapes are possible. In the study area, they can be incorporated in the existing street curb. In comparison, bio-retention areas and stormwater trees need more width. Significant land area is necessary for swales because of their shallow side slopes, and much land is in demand for larger elements like basins.

Table A3 in Appendix A shows the selected elements with their proposed hydrologic functions and main intentions when implemented in the study area. The placement criteria were derived from recommendations from international and local guidelines, as well as characteristics of the elements [27–38]. The principle was to build on existing structures as a way of retrofitting. Existing green verges in the road system were used to avoid complete unsealing. From the placement criteria, in turn, suitable spatial typologies (see Step 3) for each element were derived. The dimensions are based on the actual availability of space in the study area. However, the individual elements could also be designed in other dimensions.

### 3. Results and Discussion

*3.1. Multifunctionality—Hydrological Dimension*

The implementation of different UGIs along the whole management train (source control, conveyance, and retention or end control), as recommended in several manuals and studies [21,27,37,38], was defined as a principal strategy to improve "runoff control" in the hydrological criteria. Such small interconnected elements can counteract the risk of over-loading individual UGI elements in contrast to stand-alone UGIs [39], and this is the only means for meeting multiple design criteria [36].

Within the study area, different placement possibilities based on Table A3 in Appendix A for decentralized source-control elements were identified: permeable pavements, conveyance elements, such as infiltration trenches and bio-retention areas in the road system, and individual end-control measures in larger open spaces. However, UGI elements in the form of end-control measures placeable in larger open spaces next to the river course are expected to

be the most effective strategy for controlling runoff as a result of larger storm events. This is possible only in sub-basins 1, 2, 4, 6, and 7, as revealed in the site analysis (Step 1). In the other sub-basins, a limited amount control of the outflow is only possible with decentralized measures through source-control and conveyance elements within the road system.

Sub-basin 2 (see Figure 4), representing the largest sub-basin of the study area with a size of 9.6 ha, was chosen to demonstrate the potential of UGIs related to the hydrological dimension. The maximal implementation was approached by using the total storage capacity and potential runoff reduction as hydrological performance indicators, as recommended by Wang, Pauleit et al. [40]. Table 4 summarizes the identified UGIs suitable for sub-basin 2, including the number of elements and storage capacity. The underlying calculations of storage capacities and drainage areas for Table 4 are presented in Tables A1 and A2 in the appendix, while Figure 9 shows the potential placement of UGIs in sub-basin 2.

**Table 4.** Kind, amount, or length as well as the individual and accumulated (total) storage capacity of all identified UGIs suitable for sub-basin 2.

| UGI Element | Number or Length of Elements | Storage Capacity [m$^3$] | Total Storage Capacity [m$^3$] |
|---|---|---|---|
| Stormwater tree | 13 elements | 1.8 | 24 |
| Bio-retention area | 7 elements | 16.8 | 118 |
| Infiltration trench | 316 m | 0.27 per meter | 85 |
| Detention basin | 1 element | 1394 | 1394 |
| Constructed wetland | 1 element | 884 | 884 |
| **Total** | | | **2505** |

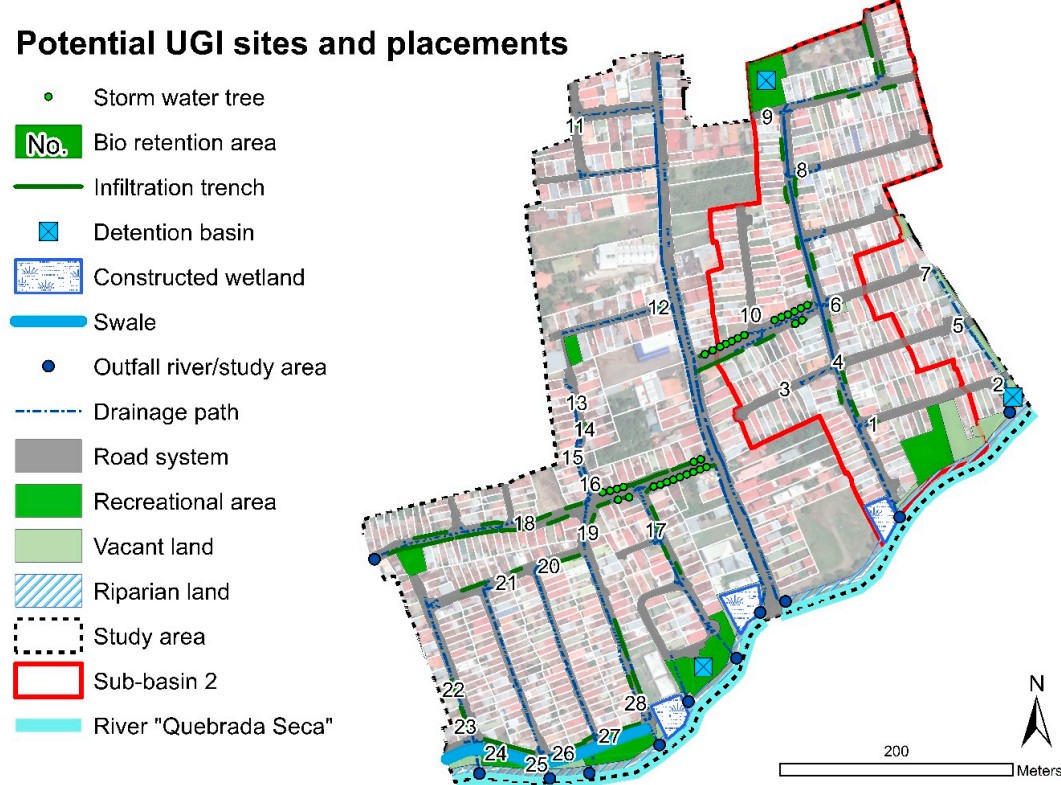

**Figure 9.** Potential UGI sites and placements for achieving the hydrological dimension of multifunctionality. Numbers 1–28 show suitable locations for bio-retention areas.

To calculate the storage capacity, no antecedent rainfall was assumed, and therefore, the soil was not saturated. The calculation leads to an expected total storage capacity of around 2505 m$^2$ for all elements. The larger UGI elements in open spaces provide the largest storage capacities (56% of the total storage capacity for the detention basin and 35% for the constructed wetland). The conveyance elements along with the road system, on the other hand, provide only a small proportion of the total capacity (9% in sum).

The potential runoff reduction for different return periods of rainfall was calculated based on respective rainfall amounts (in mm) from a study of Oreamuno et al. [41] and the size of the area (9.6 ha) to determine the volume of surface runoff. The total storage capacity of the UGI elements was then divided by the surface runoff volume to get the potential runoff reduction in terms of percentages (see Table 5).

**Table 5.** Rainfall characteristics [41] and storage potential of the UGI elements proposed for sub-basin 2.

| Rainfall's Statistical Return Period (Years) | Rainfall Amount [mm] | Runoff Rainfall [m$^3$] | Total Storage Capacity [m$^3$] | Potential Runoff Storage (%) [1] |
|---|---|---|---|---|
| 2 | 78 | 7472 | 2505 | 34 |
| 5 | 98 | 9385 | 2505 | 27 |
| 10 | 113 | 10,820 | 2505 | 23 |

[1] Performance depends not only on the storage capacity and size of drainage area, but also on the rainfall characteristics, the slope of the area as an inflow characteristic, and the type of vegetation.

Between 23% and 34% of total stormwater runoff reduction can be achieved in the example sub-basin for the rainfall events. Five different UGI elements of the management train—fulfilling functions of source control, filtration, retention/detention, conveyance, and end control—could be implemented.

Since pluvial flooding is not an issue in the study area because of the pronounced topography (the average slope is 7%), stormwater runoff reduction might be more relevant for reducing downstream fluvial flood problems. In a densely urbanized area, such as the studied one, the implementation of a combination of several retrofitted runoff reduction measures at suitable distributed sites is necessary to achieve a significant effect. This has been confirmed across other studies [20,42]. The stormwater runoff reduction described above represents the maximum hydrological potential (storage capacity) of UGIs in the sub-basin—resulting from the analysis of sub-basin 2 (Step 1) and the spatial typology (Step 3)—and, therefore, only for the hydrological dimension of multifunctionality to be considered. However, to understand the specific hydrologic response, the effectiveness and cumulative effects of the proposed UGI elements should be further investigated through modeling by using, for example, the EPA Storm Water Management Model (SWMM), a dynamic rainfall–runoff simulation model simulating both water quality and the quantity of urban stormwater runoff for single events, as well as long-term simulation of small urban catchments Modeling can be a cost-effective means to study the performance of UGI elements before implementation [15].

The management train could also be supplemented with more source-control elements at the property level, such as rainwater harvesting systems. Further exemplary possibilities are green roofs or rain gardens as alternative design options for bio-retention areas. In the road system, the possibilities of source-control elements as the first level of the management train are rather difficult. For stormwater trees as a stand-alone feature, only a few potential locations were found, and permeable pavements were only considered to be useful in the context of reconstruction measures, as unsealing on larger scales is not recommended.

*3.2. Multifunctionality—Ecological Dimension*

For the ecological dimension of the multifunctionality of UGIs in the study area, the specific design of UGI elements (biodiversity-friendly plant selection and planting, use of above-ground vegetated elements where possible, use of UGIs of higher ecological value), as well as their placement (increasing connectivity between existing green spaces, development of a continuous roadside greenery network), matters.

The integration of vegetated above-ground vegetated elements in the road system and the development of a roadside greenery network can be facilitated by all of the proposed UGI measures. All of the proposed UGI measures can be vegetated and can potentially contribute (when correctly planted and designed) to the enhancement of biodiversity. Larger elements are assumed to have higher ecological value [43]. This would be the case with bio-retention areas within the road system. The development of the roadside greenery network, when referring to trees, is, on the other hand, rather limited in terms of space in the study area. Linear elements along the road system can contribute to the ecological connection of green spaces, especially infiltration trenches. Another strategy was to select UGI elements with high expected ecological value in creating new habitat types, which are typically larger end-control elements [27,36,37,44]. Properly designed and, later, maintained, these are mainly constructed wetlands in proximity to the river in our study area. Three potential sites for constructed wetlands were identified within the study area (Figure 9). The implementation of these wetlands would create new aquatic habitats along the river corridor. After implementation, a range of biological investigations could measure the value of the ecological aspect of UGIs concerning biodiversity improvement. Important aspects are the occurrence, abundance, range, and species richness of wild pollinators and biological control agents [43].

The evaluation of improvements in the ecological multifunctionality dimension (criterion of biodiversity enhancement) due to the proposed UGI elements can, at this planning stage, be realized with regard to the relative increase of new green space, length of roadside greenery, area of above-ground vegetated elements, and areas for the new habitat types created. The results are summarized in Table 6.

**Table 6.** Evaluation of indicators of the ecological dimension of multifunctionality.

| Indicator of Ecological Multifunctionality Dimension | Achievement through Full UGI Implementation | Remarks |
|---|---|---|
| Increase of green space (excluding riparian land) [%] | 2.2 | Additional green space provided by bio-retention areas and stormwater trees |
| Length of roadside greenery activated [m] | 1500 | Infiltration trenches and stormwater trees |
| Area of above-ground vegetated elements created [m$^2$] | 3446 m$^2$ of wetland area, 2948 m$^2$ of bio-retention areas, 99.2 m$^2$ of stormwater tree pits | 3 constructed wetlands, 28 bio-retention areas, 32 stormwater trees |
| New habitat types created [-] | 3 | Constructed wetlands |

These evaluation indicators relate only to area/land cover and quantitative proxies to estimate biodiversity enhancement, and do not express the qualitative value of habitats or the quantitative increase in species or genetic diversity. During the design process, it is hard to estimate the future level of biodiversity of the proposed UGI elements. In general, the choice of planting has a great influence on the ecological value of the UGI element. Therefore, biodiversity-friendly planting is recommended among all UGI measures. Additionally, with an adapted design, the ecological value of the proposed UGI elements might even be increased. Despite this, Monberg et al. [45] developed an index score for different above-ground vegetated types of UGIs reflecting the structural heterogeneity potential and habitat provision in urban green areas to assess potential ecological benefits of different UGI designs. It was shown that an adapted "bio-design" (e.g., the employment of deadwood, terrain differences, stones, etc.) can greatly increase the ecological value of standard UGIs among all investigated elements (infiltration trenches and bio-retention areas, such as curb extensions, rain gardens, swales, dry and wet basins) [45]. The results indicate that a change in approach from the classic standard (hydrological/hydraulic) design can have a major impact on the ecological value of a UGI element.

From an ecological point of view, it could be beneficial to combine and supplement the engineered elements investigated in this work with other classical green features, like simple street green with native vegetation, community gardens, urban parks and forests, and a restored river corridor. They are principally easier and simpler to implement and require less technological expertise, but may simultaneously have a large impact on the urban biodiversity of the site. Similarly, green roofs are classified as ecologically valuable in the literature (see, e.g., Cvejic et al. [46]). There is commonly a large share of roof area in residential neighborhoods, as in the case of our study area. We limited our study to public space, as rooftops in residential areas are private properties, and implementation and promotion of UGIs, such as green roofs or rainwater harvesting, would require different policy and financing mechanisms. However, significant ecological gains are possible when considering the roof area, but roofs need to be suitable for UGI implementation.

Further benefits in addition to enhanced biodiversity and habitat creation can be expected from the UGI elements after implementation. Thus, UGI implementation can also bring further advantages for the study area. One often-occurring problem due to high degrees of urbanization concerns urban heat islands. A surface-wide implementation of UGI elements, and especially the planting of trees distributed over the surface, can counteract this and regulate the local climate [47].

### 3.3. Multifunctionality—Social Dimension

With regard to the social multifunctionality dimension, one strategy in our study area was to integrate UGIs to calm traffic with the aim of increasing safe accessibility to public (green) space. Of all considered UGI elements, bio-retention areas are the most suitable for this purpose, since sites for their implementation are not limited by vertical space requirements, as in the case of stormwater trees. In the study area, there are multiple placement opportunities for these elements within the road system, where they can be designed as curb extensions for traffic-calming (see Figure 7). In addition, these elements can be used specifically to create additional space for other uses, such as outdoor seating or where particularly safe street crossings where they are needed. Since bio-retention areas are flexible in their design, further implementations are also possible. Within the scope of this work, the elements designed as pinch points or chicanes in street design were shown as punctual area elements. A narrower linear implementation would also be possible (e.g., acting as a lane-shift element) and would provide a traffic-calming function (see Figure 10).

However, the current road design, traffic volume, and general prioritization of motorized traffic inhibit the implementation of larger and more restrictive UGI elements in the road system. If communities are willing to implement UGI elements, they can offer a great opportunity to redesign the existing traffic configurations. Municipalities could act as a role model and undertake a radical redesign by applying UGI elements. Roads could be designed to reduce traffic through spatial restrictions, such as one-way streets, and could significantly increase possibilities for UGI implementation. The following map with illustrated design examples from the US National Association of City Transportation Officials (NACTO) [48] shows an example of the possibilities in the study area for a radical redesign of the existing road system (see Figure 10). This would create possibilities to change the behavior of road users away from motorized traffic and biased towards pedestrians. A survey in the study area showed that the majority of the surveyed inhabitants welcome more green areas on the one hand, but, on the other hand, do not want a restriction of car-designated areas [48]. However, it is suggested that people adapt to their conditions; therefore, human behavior, which governs traffic engineering, is principally adaptable and not fixed through a proactive design [48].

The so-called "green streets" in Figure 10 could be designed as shared streets, which, despite allowing access for residents with motorized traffic, could be designed in a traffic-calming manner, thus providing more space for other road users, such as cyclists and pedestrians on the roads. The road lanes could be designed to be narrower, thus providing space for GI elements, wider green spaces, cycle paths, marked parking areas, etc.

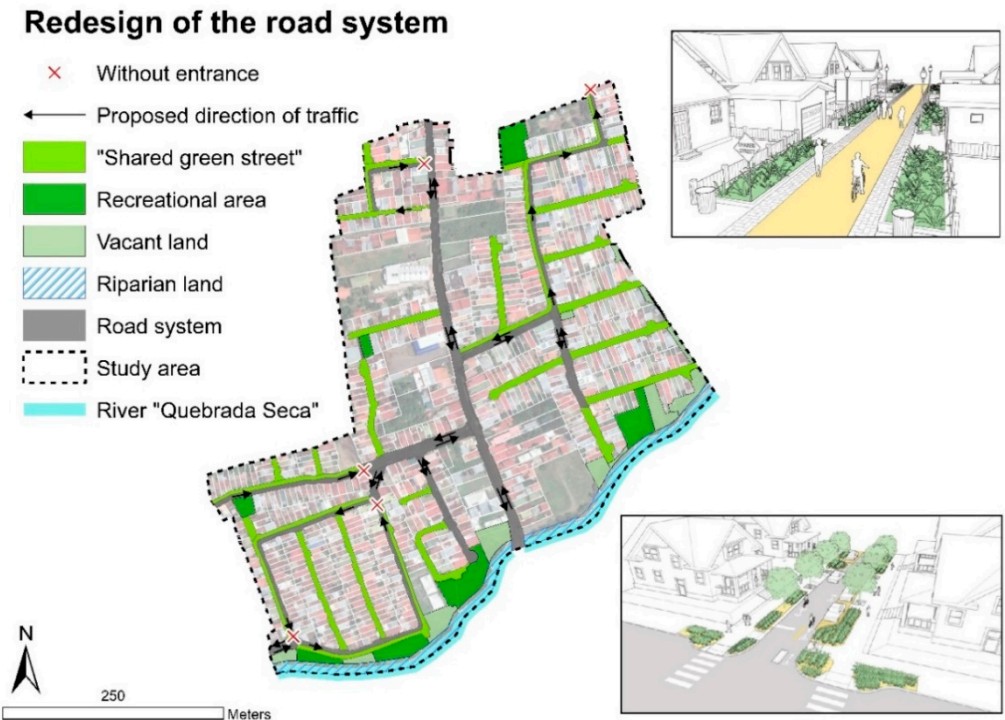

**Figure 10.** Proposal for a redesign of the road system in the study area in order to improve the social dimension of multifunctionality (source of images: https://nacto.org/publication/urban-street-design-guide/).

When thinking about a network, another strategy is to increase or provide physical connections between green spaces. In our study area, green spaces are fragmented along the river corridor due to fencing and a closely built environment. A field screening of the area showed that connecting them is difficult to achieve. However, the proposed physical connection could provide a good opportunity to integrate the constructed wetland, create links between urbanizations, and ensure a pedestrian-friendly crossing on the main road of the site.

From a social point of view, additional benefits outside of better accessibility can be expected from UGI elements after implementation. New recreational spaces in vacant land within the study area could be created, and existing recreational areas could be converted into parks for recreation for all residents. The proposed UGI elements provide an opportunity for environmental education, reconnecting people to nature, and social cohesion towards healthy urban living in the study area.

The prioritized criterion for the social functionality dimension is accessibility. It is related to the placement and design of UGIs to calm traffic for better and safer walkability, as well as to increase the physical connection to green spaces for recreation. The purpose is to encourage active travel, outdoor activities, and recreation within a neighborhood [24]. The criterion of accessibility was evaluated with the increase in length (in meters) of roads with an improved category (Table 7).

**Table 7.** Evaluation of road type change due to the proposed redesign of the road system in the study area.

| Change of Road Type | Length of Road Type Change [m] | UGI Traffic-Calming Options Gained Due to Road Type Change |
|---|---|---|
| 2 → 3 | 350 | Bio-retention area, swale, stormwater trees |
| 3 → 4 | 750 | Bio-retention area, swale, stormwater trees, permeable pavement |
| 4 → 5 | 100 | Permeable pavement |

### 3.4. Overall Discussion

The previous sections revealed that there is potential for UGI placements to achieve the different dimensions of multifunctionality. In general, the literature on UGIs related to the implementation of stormwater controls focuses on the hydrological dimension. Similarly, modeling tools primarily assess water quantity and quality criteria, but neglect assessing UGI elements related to biodiversity and social values. This is problematic, as each criterion is important to consider during the design phase. As Lähde et al. [21] also suggests, "If biodiversity criteria have failed, it has a degenerative impact on both the amenity and water quantity management potential of the site." If the delivery of multifunctional benefits is not considered during the design process, then this could ruin any chances of achieving goals related to multifunctionality [21]. Therefore, methods for the assessment of complex criteria and complete understanding of multifunctionality need further development. Decision-making and selection of adequate UGI elements are a difficult task due to the complex interactions between natural, social. and built environments. In this work, a study area was used to consider the different dimensions of multifunctionality with the selection of suitable UGI elements and the analysis of possible placements and geometries considering the spatial features of the site. The information on spatial conditions in this study area is detailed; therefore, a detailed investigation could also be undertaken in this case. Nevertheless, the procedure of analyzing the space availability of a study area regarding existing traffic and green spaces can generally be adapted to other similar areas—even with less data—for sustainable urban transitions. The expected multifunctional performance of UGIs as a basis for the selection of particular UGI elements in this study (see Table 3) is highly dependent on the design of the elements and subsequent maintenance measures.

The analysis of spatial typology (road system and open space characterization as well as site-specific constraints, e.g., driveways), as proposed in our methodological approach, is practical, and the establishment of multifunctional design criteria and related implementation strategies can reveal the placement potential for UGIs and provide guidance for efficient implementation at the neighborhood level. Through the use of quantitative indicators for different design criteria, the potential impact can be estimated and compared to traditional/conventional (gray) solutions. Thinking beyond hydrological functioning of UGIs can lead to significant gains in multifunctionality. In particular, beneficial on-site functionalities of strategic placement of multifunctional UGIs can deliver additional arguments to the often off-site (downstream) perceived benefits of runoff control.

When looking at specific UGI elements, in principle, bio-retention areas can be placed as traffic-regulating elements at many locations in the study area. The infiltration trench element offers great possibilities if the road design is not to be changed. For constructed wetlands, three suitable sites were found in the study area, which, as end-control elements, could offer a certain retention even during larger storm events for the respective sub-basins, and could further be used for graywater treatment.

The suitable open spaces for implementing GI elements considered in the current analysis correspond to 10% of the total area. Basically, the implementation options are always limited to the space availability onsite. Further opportunities can arise if more usable areas are created, for example, through the conversion of undeveloped properties, where the community can reach an agreement with landowners. Undeveloped properties account for 7% of the total area. This would also provide more opportunities for larger conveyance elements for more runoff control onsite in the study area, and not just large-scale elements along the river, which would always act as an end-of-pipe solution.

In general, more possibilities should be exploited by implementing elements in the road system, as 16% of the total area in the study area correspondents to roads. UGI elements offer many multifunctional possibilities here. Even if the hydrological benefits are limited to the retention and reduction of runoff directly falling on the element (source control) or to the first flush, then at least smaller rainfall events can be controlled. In the ecological and social dimensions, greater benefits can be expected. The green network with individual decentral-

ized measures can be developed on a smaller scale, which contributes to the biodiversity onsite, and correctly placed UGI elements can function as traffic-regulating elements, thereby changing the traffic behavior of residents, providing more safety and accessibility in the area, aesthetically enhancing the space, and counteracting the heat island effect. The analysis shows that many different measures can be integrated into the road system, regardless of whether the street design should be changed and its traffic restricted, or if only the current conditions are used; therefore, stormwater trees, bio-retention areas, and infiltration trenches are suggested as the most suitable ones. The potential for implementing UGI measures is given for the whole road hierarchy and all traffic volumes throughout the study area. The many existing green verges in the road system can be used well for implementing UGI elements there; thus, the runoff is not just passed by, as it is at present. As a result, additional unsealing is reduced or not necessary at all.

Due to the flexible design of bio-retention areas, this type of UGI provides several possibilities. These can be designed as aerial or linear elements or as transport elements in the road system, but also, in principle, on properties themselves or in open spaces as larger (end-control) elements. Correctly placed, the elements could be used as a traffic-calming technology in the road design. Here, many different configurations and further treatment of graywater are possible.

Multifunctionality should not be neglected when selecting the right UGI elements and their locations. The hydrological dimension should not be considered alone, as it typically is in the literature. The present work could show that UGI elements can simultaneously offer many other positive dimensions for urban areas. UGI elements provide an opportunity to reconsider the existing road design and create more recreational areas for residents. In addition, properly designed UGI elements can contribute to urban biodiversity and create habitats.

In principle, a combination of different measures in the management train is recommended. This is suitable not only in the hydrological sense, but also in the ecological one. Vegetated UGI elements, as smaller green spaces decentralized on properties and distributed over the road system, together with larger green areas and particularly ecologically valuable elements on them, result in a woven green network over the entire area, which can have a strong positive effect on the urban biodiversity of the site.

A combination is also recommended in the social sense. Voluntary commitment and participation of residents in UGI promotion can be achieved at the property level, for example, when people choose on their own to collect rainwater in Rain Water Harvesting (RWH) systems. In the road system, the elements can contribute to, among other things, the safety of residents and, in larger areas, to more recreational areas. However, this work did not consider the possibilities of UGIs on properties themselves. Nevertheless, in the literature, especially for densely urbanized areas, the opportunities for UGI implementation on properties themselves are estimated as valuable. Often, green roofs are recommended, and are also considered useful in terms of their ecological benefits. According to this, the use of rainwater by RWH systems, as well as further UGI elements (e.g., rain gardens) as prevention measures and the first stage of the management train, could be promoted more, as was also undertaken by the Municipality of Curridabat, Costa Rica for small-scale UGIs on private lots [49]. In the case of green roofs, however, particular building specifications are required to support the additional weight from the plants and substrates; therefore, they are probably more likely to be implemented in future building projects, rather than in existing ones as retrofitting measures.

The present work showed that retrofitting is often difficult and is associated with many uncertainties and restrictions in planning. Extensive unsealing must often take place, and major changes to the street design could lead to later conflicts with the residents. In contrast, implementation is considered easier, cheaper, and more probable in the case of reconstruction measures. We recommend that municipalities consider multifunctional UGI elements in the planning of new urban areas. Permeable surfaces could be used right from the start as pavements in areas with low traffic volumes, as well as stormwater trees for

temperature regulation, bio-retention areas for controlling traffic in a targeted manner, and larger collective storage facilities to create green spaces and urban parks. In the case of the municipality of Flores, such considerations could be applied to the expansion of the residential area to the "undeveloped land—east" (see Figure 3). Simultaneously, legal principles should be created, and policy instruments—depending on the responsibility—such as the "plan regulador" in Costa Rica should be used.

### 3.5. Our Methodology in the Context of Other Recent Studies

Kuller et al. (2019) developed and tested a placement tool for UGIs for Australia, but it is focused on hydrological functioning and does not consider the specific constraints involved in urban road design for less-developed countries [50]. Giacomoni and Joseph (2017) conducted a study on UGI placement optimization, but also with a narrow hydrological focus [51]. A study by Hasala et al. (2020) presents an approach to participatory mapping to assess local perceptions of nuisance flooding [52]. Their approach goes beyond hydrological design principles and includes residents' perceptions of potential benefits. However, multifunctionality in UGI placement, as in our methodology, is not considered.

Tran et al. (2020) introduced the Green Infrastructure Space and Traits (GIST) model as a tool for evaluating and maximizing UGI multifunctionality on a city-wide scale based on optimizing site selection and plant traits in UGI design [53]. Their derivation of multiple benefits based on plant characteristics could be potentially applied in our spatially explicit neighborhood-scale methodology. This should be considered in future research investigations.

For a Global South context, in their study, Nugroho et al. [54] determined the placement of UGIs in accordance with the criteria of land suitability and analyzed the effectiveness of their application for an urban catchment in Jakarta, Indonesia [53]. Bio-retention and rain barrels were identified as suitable UGI elements, and their effectiveness in runoff reduction could be proved by modeling. However, other functional dimensions, such as social or ecological dimensions, were not assessed.

With regard to the functionality of micro-climate regulation, Norton et al. [47] presented a framework that enables prioritization of placement and UGI type at the neighborhood scale. Since this functionality of the social dimension was not considered in our study, it could be promising to integrate or combine their and our placement methodologies to represent another form of UGI multifunctionality.

## 4. Conclusions

Urbanization has increased the volume and peak flows of stormwater runoff, exposing communities to greater risks of floods. In the future, climate change will further exacerbate the consequences of storm events. The development of UGIs, such as stormwater retention facilities, the reduction of impervious surfaces, and the promotion of green spaces, can mitigate several of the negative consequences of both. Such an approach can also enhance the ecological function of urban spaces and bring many social benefits in promoting sustainable living. UGI elements provide multiple beneficial functions for territorial development within the same spatial area. This study investigated the potential of multifunctional UGI placement at the neighborhood scale for a tropical country. For this purpose, an exemplary area was investigated, suitable multifunctional design criteria and strategies were defined, and, on the basis of the available space, spatial possibilities for various suitable UGI elements regarding their placement, spatial distribution, and geometries were investigated. Thereby, the possibilities for UGI elements in relation to the characteristics of the study area were shown by means of a general methodology. Nevertheless, when it comes to actual implementation, further investigations in the field and modeling are essential. Different UGI elements should be combined to not only provide better control of all levels of the stormwater management train, or of an entire spectrum of storm events, but also to exploit the full range of potential ecological and social benefits.

The proposed methodology can guide local planners and decision-makers in UGI implementation at the neighborhood scale. Through an analysis of local preconditions and with site-specific design criteria, the multifunctional benefits from UGIs could be further amplified. If UGI elements are to be successfully implemented in a densely urbanized area, then careful coordination of the interdependences of the different aspects of multi-functionality and other urban functions in the design process is essential; considering all dimensions of multifunctionality requires a structured and strategic approach.

Therefore, one possibility for the future would be to design a multi-criteria tool to assist in the screening and selection of measures adapted to the climatic conditions in tropical countries of the Global South. However, it is questionable to what extent further multifunctional advantages of UGIs can be taken into account. The development of UGI design guidelines adapted to the respective data situations onsite and the local conditions is considered to be useful, as such guidelines currently only exist for developed countries. The development of such guidelines would enable local difficulties, such as on-street, untreated graywater runoff or the demand for low-cost UGI solutions, to be incorporated into the design specifications of GI elements.

All in all, many ecological and social problems of urban areas in tropical countries of the Global South could possibly be counteracted with engineered UGI elements: Existing drainage systems can be relieved, the problem of flooding countered, the sealing of the area reduced, public green recreation areas created, the heat island effect reduced, other traffic participants (like pedestrians) supported, and untreated graywater purified. Thus, the concept of UGI offers many possibilities far beyond the hydrological aspect for both industrialized and developed countries. Finally, the involvement of local responsibility and the population is required. Social acceptance, creation of local knowledge, and an available budget are essential basic requirements for successful UGI projects. Thus, UGI elements may become an integral part of spatial planning and territorial development, as they offer a better alternative to standard gray solutions by providing multiple functions.

**Author Contributions:** Conceptualization, T.F., F.C., and J.H.; methodology, T.F., F.C., and J.H.; software, T.F.; validation, T.F. and J.H.; formal analysis, T.F. and J.H.; investigation, T.F., F.C., and J.H.; resources, T.F., F.C., and J.H.; data curation, T.F.; writing—original draft preparation, T.F. and J.H.; writing—review and editing, T.F., F.C., and J.H.; visualization, T.F.; supervision, J.H.; project administration, J.H.; funding acquisition, J.H. All authors have read and agreed to the published version of the manuscript.

**Funding:** This research was funded by the German Federal Ministry of Education and research (BMBF), grant number 01UU1704.

**Institutional Review Board Statement:** Not applicable.

**Informed Consent Statement:** Not applicable.

**Data Availability Statement:** All data used in this study was presented or referenced in the manuscript.

**Acknowledgments:** We acknowledge the support from the German Research Foundation (DFG) and the Open Access Publishing Fund of the Technical University of Darmstadt.

**Conflicts of Interest:** The authors declare no conflict of interest.

## Appendix A

**Table A1.** Overview of storage capacity calculations for the considered UGIs.

| UGI Element | Formula | Calculation | Average Storage Capacity |
|---|---|---|---|
| Stormwater tree | Area per element [m$^2$] * ((Depth of system [m]* Void volume of filled substrate mixture [%]) + ponding depth [m]) | (1.75 m * 1.75 m) * ((1.0 m * 0.5) + 0.1 m) | 1.8 m$^3$ |
| Bio-retention area | Area per element [m$^2$] * ((Depth of soil [m] * Void volume soil [%]) + (Depth of gravel [m] * Void volume gravel [%]) + depth of ponding [m]) | 30 m$^2$ * ((0.2 m * 0.7) + (0.8 m * 0.4) + 0.1 m) | 16.8 m$^3$ |
| Infiltration trench | Average area per element [m$^2$] * ((Depth of soil layer [m] * Void volume soil [%]) + (Depth of gravel layer [m] * Void volume gravel [%])) | (0.5 m * 27.8 m) * ((0.2 m * 0.7) + (1.0 m * 0.4)) = 7.5 m$^3$ | 7.4 m$^3$ (=0.27 m$^3$ per meter) |
| Detention basin | Available open space [m$^2$] * Average depth of element [m] | 1245 m$^2$ * 1.12 m | 1394 m$^3$ |
| Constructed wetland | Available open space [m$^2$] * Average depth of element [m] | 850 m$^2$ * 1.04 m | 884 m$^3$ |

Calculated with dimensions from Table A2; soil and gravel layer depth according to [55].

**Table A2.** Overview of drainage area calculations for the considered UGIs.

| UGI Element | Formula | Calculation | Average Drainage Area |
|---|---|---|---|
| Stormwater tree | Average flow length [m] * Average width of drainage area [m] | 7.8 m * 25.5 m | 197.6 m$^2$ |
| Bio-retention area | Average flow length [m] * Average width of drainage area [m] | 46.2 m * 23.5 m | 1085.7 m$^2$ |
| Infiltration trench | Average flow length [m] * Average width of drainage area [m] | 35.5 m * 23.5 m | 834.3 m$^2$ |
| Detention basin | Entire drainage area upstream from placement site | - | 1.6 ha |
| Constructed wetland | Entire drainage area upstream from placement site = Sub-basin area | - | 9.6 ha |

**Table A3.** Overview of the UGI elements and their characteristics.

| | UGI Element | Hydrologic Functions | Main Intention | Placement Criteria | Suitable Spatial Typology | Dimension Restrictions |
|---|---|---|---|---|---|---|
| Road system | Permeable pavement | Source control Conveyance control Filtration, infiltration | Multi-use in dense areas Stormwater infiltration | Low traffic volume | Road types 4 and 5 | none |
| | Stormwater tree | Source control Retention Filtration | Development of roadside greenery network Buffer to road traffic | Buildings: Min. 2 m distance [36] Green verges: Must be given Driveways: Street sections excluded and min. 2 m distance [36] Intersection/corner: Min. 3 m distance [36] Roadside greenery: 6 m distance in between [36] | Road types 2, 3, and 4 | Length: 1.75 m Width: 1.75 m Depth: 1 m (+0.1 m ponding) |
| | Bio-retention area | Conveyance control Retention Filtration | Vegetated UGI in road system Traffic-regulating element | Green verges: Min. 10 m length Driveways: Street sections excluded Sewer inlet or drainage path: Proximity Flow length: Max. 100 m | Road types 3 and 4 | Length: max. 10 m Width: 3 m Depth: 1 m (+0.1 m ponding) |
| | Swale | Conveyance control Retention Filtration Infiltration | Vegetated UGI in road system | Slope: 0, 5–6% recommended [27,30] Green verges: Min. 5 m length Drainage area: Max. 2 ha recommended [30] Roadside greenery: Difficult to integrate Greywater outlets: Street sections excluded Driveways: Street sections excluded | Road types 3 and 4 | Length: min. 5 m Width: 2.5 m Depth: 0.5 m |
| | Infiltration trench | Conveyance control Infiltration | Vegetated UGI in road system | Green verges: Min. 10 m length | Road types 2 and 3 | Length: min. 10 m Width: 0.5 m Depth: 1.2 m |
| Open space | Detention basin | Detention Conveyance control/end control | UGI effective for larger events UGI with high potential ecological value Green space enhancement and conversion | Size of drainage area: Max. 2 ha [30] Size of open space: Min. 1/8 of drainage area | Recreational area Vacant land | Depending on the design goal and available space |

**Table 3.** *Cont.*

| UGI Element | Hydrologic Functions | Main Intention | Placement Criteria | Suitable Spatial Typology | Dimension Restrictions |
|---|---|---|---|---|---|
| Constructed wetland/ Retention basin | Retention Filtration Infiltration End control | UGI effective for larger events UGI with high potential ecological value Green space enhancement and conversion | Size of drainage area: Min. 5 ha [36] Available open space: Next to outlet river | Riparian land Vacant land | Depending on the design goal and available space |

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
