# Peer review of "A Methodology for Assessing the Implementation Potential for Retrofitted and Multifunctional Urban Green Infrastructure in Public Areas of the Global South"

_sustainability, doi:10.3390/su13010384_

Round 1

Reviewer 1 Report

Overview

The manuscript deals with the importance of Urban Green Infrastructures (UGIs) in the territorial planning of urban (residential) areas in the Global South using certain methodology for assess their implementation potential and highlighting their multifunctional aspect, in this case, hydrological, ecological and social.

General comments

The writing style of the manuscript is clear, concise, precise, fluid and understandable, which helps to capture the reader's interest. The methodology of evaluation proposed is simple and explained in detail. The four steps that the authors suggest to follow for the implementation of UGIs are reasonably justified and follow a logic of spatial and temporal planning.

In general terms, it can be considered a work of interest to urban planners and managers and its relevance lies in highlighting the need to move towards a new form of urban planning that is more environmentally friendly and sustainable. The manuscript brings little scientific novelty and, in my view, does not fully demonstrate the specificity of this methodology for urban contexts in developing countries or neighbourhoods of these countries (this is the scale of the methodology). On many occasions, approaches or methodologies that are perfectly applicable in the developed world fail to be transferred without an adaptive vision to the developing world. It is likely that if the general discussion or conclusions of the manuscript insist on highlighting the viability of retrofitted UGIs for specific cases in the Global South, the work would improve in its scope and impact.

Considering these improvements that I suggest, I think the work is a very good assessment of the potential of IGU implementation and, agreeing with the authors, UGI elements may become an integral part of spatial planning and territorial development, as they offer a better alternative to standard grey solutions by providing multiples functions.

Some specific comments

Lines 81-94: This is a summary of the study, anticipating results and conclusions. It is not specifically a problem but it is unusual to find it at this location in the manuscript.

Line 160: Please explain what are the unsealed lands? Is it public land that can be built on?

Line 318 (Fig. 3): Clarify the qualitative classification of the potential multifunctional performance of land use. The differences among swale, bio retention area, infiltration trench, detention basin and retention basin are not very clear. In this table or in the text the differences should be made explicit and examples could even be mentioned.

Line 368: What do the numbers in the Figure 9 (1 to 28) mean? Please, clarify.

Line 387: Exactly, what data does column 1 of Table 5 provide? What does TR mean?

Author Response

Dear reviewer,

thank you very much for taking the time to review our manuscript. We are very pleased by your generally positive feedback and we are thankful for your suggestions for improvement. 

This is how we addressed your suggestions and comments:

  • Lines 81-94: This is a summary of the study, anticipating results and conclusions. It is not specifically a problem but it is unusual to find it at this location in the manuscript.

Lines 81-94 were eliminated.

  • Line 160: Please explain what are the unsealed lands? Is it public land that can be built on?

The sentence “The latter represent areas that belong to the Municipality and are reserved for future development.” was added in line 160 to explain what is meant by “unsealed areas”.

  • Line 318 (Fig. 3): Clarify the qualitative classification of the potential multifunctional performance of land use. The differences among swale, bio retention area, infiltration trench, detention basin and retention basin are not very clear. In this table or in the text the differences should be made explicit and examples could even be mentioned.

The differences among swale, bio retention area, infiltration trench, detention basin and retention basin are explained in the paragraph starting from line 321. The table (caption) makes reference to the qualitative classification’s underlying literature and examples i.e. descriptions of the differences are given in lines 321-334.

  • Line 368: What do the numbers in the Figure 9 (1 to 28) mean? Please, clarify.

The sentence “Numbers 1-28 show suitable locations for bio retention areas.” was added to the caption of Figure 9 to give an explanation for the numbers 1-28.

  • Line 387: Exactly, what data does column 1 of Table 5 provide? What does TR mean?

The heading of column 1 of Table 5 was changed to “Rainfall’s statistical TR return period (years)” to explain its meaning.

Best regards on behalf of the authors,

Jochen Hack

Reviewer 2 Report

The contribution is particularly interesting since it addresses an important issue concerning how to combine “ecological and social benefits” through nature-based solutions.
The structure is well organized and offers interesting insights in combining the three dimensions (Hydrological, Ecological and Social) related to suitable indicators. Nevertheless, the case study does not offer a wide range of “situations” that, in other contexts, may make more complex the application and need for further indicators. The link, for instance, with the urban mobility issue, is not explained enough. The risk could be that the strategy serves single urban areas without considering the effects on the surrounding. The so-called Urban Green Infrastructure, however, is not a new approach. The urban Greenways are part of urban planning solutions for many years (decades). Therefore, I will suggest better explaining the innovative approach that the UGI is supposed to implement, correlated with the Greenways. Another suggestion regards the concept of Nature-Based solution. The concept embraces many approaches that come from technological solutions to urban design and planning strategies (resilience, thereby). I suggest better clarifying which kind of connection between UGI and NBs is proposed.

Author Response

Dear reviewer,

thank you very much for taking the time to review our manuscript. We are very pleased by your positive feedback. 

We followed your suggestions to better explain the innovative character of UGI implementation (lines 81-84 in the revised manuscript) and to stress more the connection between UGI and NBS. The latter has been clarified with additions to lines 55 and 57. In line with the IUCN definition of NbS, Green Infrastructure (or Urban Green Infrastructure as we termed it) is a specific subset of NbS.

Best regards on behalf of the authors,

Jochen Hack

Reviewer 3 Report

The topic of the manuscript is very interesting and timely, however, it needs a minor revision to meet the standards of Sustainability. Here are my comments for improving the paper:

1- In the introduction section, make the knowledge gaps and research objectives clearer.

2- Line 62-71, since you mentioned that, you can read more papers in this field and use them in this part. Such as:

  • Buijs, A. E., Mattijssen, T. J. M., Jagt, A. P. N. Van Der, Ambrose-oji, B., Andersson, E., Elands, B. H. M., & Møller, M. S. (2017). ScienceDirect Active citizenship for urban green infrastructure : fostering the diversity and dynamics of citizen contributions through mosaic governance. Current Opinion in Environmental Sustainability, 22(June 2016),
  • Faivre, N., Fritz, M., Freitas, T., & Boissezon, B. De. (2017). Nature-Based Solutions in the EU : Innovating with nature to address social , economic and environmental challenges
  • Aram, F., Solgi, E., & Holden, G. (2019). The role of green spaces in increasing social interactions in neighborhoods with periodic markets. Habitat International, 84, 24–32.
  1. The conclusion is too wordy, please polish it.
  2. Please proofread the paper as there are several writing issues.

Author Response

Dear reviewer,

thank you very much for taking the time to review our manuscript. We are very pleased by your generally positive feedback and we are thankful for your suggestions for improvement. 

This is how we followed your suggestions i.e. addressed your comments:

  • The knowledge gaps are stated in the introduction in lines 63-67 and the specific research objectives of our study are clarified in lines 73-77 of the introduction.
  • The addtional references suggested were added to the following lines: Buijs et al. (line 68-70), Faivre et al. (line 38), and Aram et al. (lines 64-67). 
  • The conclusions were revised and rephrased in several parts (lines 670, 674-676, and 679-682.
  • The paper has been proofread again by a native English speaker to improve splelling and grammar (see changes in the manuscript).

Best regards on behalf of the authors,

Jochen Hack